# Gas concentration mapping and source localization for environmental monitoring through unmanned aerial systems using model-free reinforcement learning agents

**Anees ul Husnain[1,2], Norrima Mokhtar[1]***, Noraisyah Binti Mohamed Shah[1], Mahidzal Bin Dahari[1], Amirul Asyhraff Azmi[1], Masahiro Iwahashi[3]**

**1** Department of Electrical Engineering, University of Malaya, Kuala Lumpur, Malaysia, **2** Department of Computer Systems Engineering, Faculty of Engineering, The Islamia University of Bahawalpur, Bahawalpur, Pakistan, **3** Information, Telecommunication and Control System Group, Nagaoka University of Technology, Niigata, Japan

\* norrimamokhtar@um.edu.my

**Data Availability Statement:** All relevant data are within the manuscript and its Supporting Information files.

## Abstract

There are three primary objectives of this work; first: to establish a gas concentration map; second: to estimate the point of emission of the gas; and third: to generate a path from any location to the point of emission for UAVs or UGVs. A mountable array of MOX sensors was developed so that the angles and distances among the sensors, alongside sensors data, were utilized to identify the influx of gas plumes. Gas dispersion experiments under indoor conditions were conducted to train machine learning algorithms to collect data at numerous locations and angles. Taguchi's orthogonal arrays for experiment design were used to identify the gas dispersion locations. For the second objective, the data collected after pre-processing was used to train an off-policy, model-free reinforcement learning agent with a Q-learning policy. After finishing the training from the training data set, Q-learning produces a table called the Q-table. The Q-table contains state-action pairs that generate an autonomous path from any point to the source from the testing dataset. The entire process is carried out in an obstacle-free environment, and the whole scheme is designed to be conducted in three modes: search, track, and localize. The hyperparameter combinations of the RL agent were evaluated through trial-and-error technique and it was found that ε = 0.9, γ = 0.9 and α = 0.9 was the fastest path generating combination that took 1258.88 seconds for training and 6.2 milliseconds for path generation. Out of 31 unseen scenarios, the trained RL agent generated successful paths for all the 31 scenarios, however, the UAV was able to reach successfully on the gas source in 23 scenarios, producing a success rate of 74.19%. The results paved the way for using reinforcement learning techniques to be used as autonomous path generation of unmanned systems alongside the need to explore and improve the accuracy of the reported results as future works.

**Funding:** The study was supported by IIRG003(b)-19IISS, University of Malaya, Malaysia. The grant number has also been updated in the Editorial Manager portal. The funders had no role in study design, data collection and analysis, decision to publish, or preparation of the manuscript. The funder helped in availing the proof reading services and cost of publications for this work.

**Competing interests:** The authors have declared that no competing interests exist.

## Introduction

All living creatures share the earth and its climate. Any devastating effect on mother nature disturbs the sustainability of various forms of life on it. The United Nations (UN) has consistently declared 'Climate Change' as one of the greatest challenges of these times that has threatened the survival of a significant proportion of the human population. Catastrophic floods, rising sea levels, and unprecedented weather patterns are not the only havoc that climate change brings. Indirectly, climate fuels other crises, including contaminated drinking waters, lowered crop production, climate inequality, regional health crisis, and eventually, the children. The future seems gray without effective measures taken in time to mitigate the reasons behind climate change.

According to the data released by the US Environmental Protection Agencies (EPA), Green House Gases (GHGs) emissions are the most significant contributors to global warming through energy, agriculture, and industrial sectors [1]. The need for autonomous and automated solutions becomes inevitable for governments to adapt global, regional, or local environmental protection policies. Additionally, the core component of any such solution would be the acquisition of emission data from the source point and dispersion in the environment so that the emission policies can be enforced. Although existing technologies mostly rely on fixed-position or hand-held sensors to monitor such emissions, the role of Artificial Intelligence (AI) and Robotics is auspicious.

Robotic Olfaction is an emerging area of research where the objective is to develop sensory modules for robots that can help them distinguish and classify various gases, track the gas plume movements, and localize the emission point [2]. Aerial or ground robots with sniffing abilities carry an immense potential to serve diverse areas of applied research. The same can be idealized to autonomously sense and localize fugitive emissions for aerial or ground robots equipped with AI and Machine Learning (ML) technologies.

Among the unmanned systems, aerial robots like Unmanned Aerial Vehicles (UAVs), inter-changeably called drones, have a natural advantage in sensing and tracking gases to their points of emission. Cooperative search and tracking algorithms can help multiple drones to localize emissions in larger areas with the ability to distinguish and localize a particular hazardous gas. This application has been coined Gas Source Localization (GSL) in literature [3–5]. Autonomous UAVs generally refer to their autonomous path planning ability and data collection operations simultaneously. However, considering a UAV for a GSL mission means dealing with a constantly changing environment (nature of gas molecules and wind dynamics), requiring an out-of-the-box approach. Moreover, the selection of AI technology for path planning, modeling, or simulation of a gas concentration environment and the effectiveness of onboard sensing technologies are the prime focus of this work.

This work primarily focuses on a UAV's ability to track and localize gas emissions that may be used for autonomous environmental monitoring. The framework implemented here calculates the path to the source in fractions of seconds at run-time. This work can engage various gas distribution scenarios, multiple policies, and diverse robots as future endeavors.

### The challenges

One of the fundamental challenges is the random nature of gas molecules and the acquisition of gas data at various locations through an onboard sensor module. When a gas plume reaches a contact sensor, the particles in random motions and collisions may lead toward random readings, particularly when sensors get closer to the emission point. The molecular mass, storage pressure, environmental temperature, emission point structure, conditions like wind and height of dispersion, and any combinations of such changes can affect the gas concentration

map. Therefore, to overcome these fickle factors, the 'efficacy of the sensory module' under known environmental conditions (like, wind vectors, temperature, and humidity) substantially impacts the overall results. If the sensory module does not consider the wind dynamics, then the data recorded by such a module may be useless for plume source estimation.

The second crucial challenge is ensuring the sensory module's performance onboard a UAV, as the air thrust from the UAV's propellors can directly affect the sensor readings. The air thrust from the UAV's propellors influences the gas sensor readings for contact-based gas detection sensors. It can affect the number of gas particles per million (ppm) in contact with the sensor, which could impact the natural distribution of gas molecules in the environment. Moreover, if the gas concentration indicates proximity to the source, the air thrust from the propellors will also affect the source estimation.

Third, obtain a gas concentration map from the acquired data and generate the path autonomously to the emission point. This path generation process must be on a run-time basis so that the impact of the plume motion should least affect the estimation of the source point. An increased computation time for the path generation would be ineffective for a moving gas plume source, and the algorithm may get trapped in a localmaxima with a repetitive generation of source estimate.

Finally, the generated path is validated by feeding unknown scenarios to the algorithm and verifying the generated trajectories. This work aims to overcome challenges that hinder or affect autonomous gas source localization through aerial robots or UAVs.

## Related work

### Environmental monitoring through UAVs

UAVs have a natural advantage in regulating fugitive emissions as both are airborne compared to ground-based, fixed position, or hand-held sensors. A quite useful review study published on the techniques and technologies available for monitoring the gas pipelines and leakage detection also confers the usefulness of use of drones as well [6]. Yet, using UAVs to detect and localize fugitive emissions comes up with challenges. These may include drone height, distance from the emission point, structural congestion in the environment of gas leakage, and wind impacts between the time of gas emission and sensor perception. The idea of a laser-based methane leakage detector was effective for being mounted on UAVs for gas leakage detection at a point of emission [7]. However, despite accurate localization, it has a drawback in that the laser or similar technologies can only identify single-line vertical columns directly underneath the sensor position. These approaches limit the autonomous search and track capabilities. Yet, they are handy for detecting gas leakages from miles-long stretched pipelines where the target of leakage detection is already known or probable. The onboard sensors play a decisive role in gas source localization as gas plumes are in continuous motion, and sensors generate readings when gas molecules come in contact with them [8]. The count of propellors play a fundamental role in the stability of rotor-based UAVs. The higher the number of propellors are, the more stable the flight it takes. A quite recent and notable work upon the stability of the rotor-based UAV was carried by [9], in which a control architecture was designed and implemented for a quadrotor losing its propellors and still being able to maneuver itself within the stable region.

Nevertheless, there is another problem with using contact-based sensors on UAVs. The air thrust from the propellers of UAVs affects the gas concentrations above and below it, and hence, no matter how accurate the sensing module is, the gas data will always be inconsistent. After going through dozens of studies on this problem, a niche was intuited to explore further the literature, *i.e.*, to identify the most suitable distance from a UAV's propellor to place a gas

sensor. A study was published in which numerous gas dispersion experiments were conducted with a contact-based gas sensor placed at different distances from the propeller. It was learned that the most suitable distance is 1 to 1.2 meters [10]. The viability of using UAVs and MOX sensors for gas source localization was explored further to confirm. However, in most of the studies, the impact of air thrust on the gas sensors was either overlooked or held a negligible significance in their objectives with the onboard Metal Oxide (MOX) [11, 12].

Another critical consideration is that the impact of disconnected patches in sensor reading reduces the overall localization accuracy [13]. From disconnected patches, it is preferred that during the motion of the gas sensor module, there comes a break or pause in sensor readings, and the search process must be re-initiated. All the factors mentioned above played a crucial role in designing a sensory module comprising an array of MOX sensors, presented in detail under the Air Quality Sensor Array (AQSA) section.

## Platforms for UAV's flight control and simulation

Since the attainment of confidence in using UAVs equipped with MOX sensors, this subsection explores the technology selection. Importantly, to choose between open-source and closed-source. Closed-source technology indeed comes with better implementation of standards, but still, there are a few significant disadvantages. These include the inaccessibility of the source codes, higher dependence on customer services for the slightest to any irreversible issue, and, more importantly, the high cost of it as a product in the market. At the same time, open-source technologies provide free access to coding repositories since they are free from commercial interests. However, the cost is paid in another manner: the variations in coding practices and standards across the globe. Since these differences naturally come from the open-source developer's community in the form of code dependencies. These dependencies, at times, may become troublesome to resolve. Nevertheless, open-source technologies give access to futuristic resources and peer support networks with nothing to pay for. Despite all these facts, the findings from literature considering the problem at hand are briefly presented.

Simulink-based test beds have been developed where commercial drones can be modeled and simulated alongside their environments to generate a control strategy or a trajectory [14]. Flight control models can be developed and simulated in MATLAB Simulink. One such approach was to develop an 'attitude controller' of a quadrotor that generated a high-level code in C/C++. The codes can also be deployed for open-source hardware platforms like PIX-HAWK autopilot [15].

On the other hand, a comparison was published to highlight options under open-source technologies to assemble a UAV for academic research. It included flight controllers, propulsion systems, inertial measurement units (IMUs), Global Navigation Satellite Systems (GNSSs), Barometers, Radio Control (RC) transmitters, and flight controller software to operate the unmanned system [16]. Among hardware, computing resources like Raspberry Pie (RPI), flight controllers like Pixhawk, and micro-controllers like Arduino have effectively reduced the cost of educational UAVs [17]. Recent work has integrated RPI-based drones through android applications via the cloud to perform autonomous image processing capabilities with search and track features [18].

It was learned that the Robot Operating System (ROS) and MATLAB were the top two contenders for using UAV's control programming, respectively. However, ROS is integrated with Gazebo, where one can import and customize a simulation environment, alongside RVIZ, where different forms of data from the environment or robots can be visualized. ROS runs as a core that facilitates ROS services, messages, packages, and dependencies. ROS allows simultaneous simulation of flight dynamics, onboard sensors, and flight environments, with no

change of codes, while shifting simulations to hardware and ensuring rapid proto-typing [19]. ROS has also been used to simulate multiple UAVs, supports communications among hetero-geneous robots [18], and is an ultimate choice.

## Modeling the gas dispersion phenomena

From mathematical modeling, it is inferred at large as a set of equations to anticipate the over-all change in a system by changing the variables at desired times and conditions. The same is valid for model gas dispersion phenomena; however, there are critical considerations for the nature of the model itself, affecting the model selection process. For example, a point source of gas emissions is the emissions from stacks, nozzles, pressure valves, etc. Similarly, road emis-sions are considered line sources, and treatment ponds emissions are considered area sources. The emissions are usually modeled for either short-term or long-term, ranging from a few minutes or a day to a few days to years. Metrological conditions, properties of gas particles, storage conditions, the structure of the emission point, the direction of emission, and the height of emission sources also affect the plume movement.

The situation under consideration to map gas concentration is the leakage of heavier gases that tend to stay low, stored under pressure, and dispersed in an indoor environment with an almost constant flow of wind from a point-source emission structure.

Multiple dispersion models and their simulations are available to study and better under-stand the gas distribution process. The Gaussian Plume model is an ideal deterministic approach that estimates the change in gas concentration levels with the distance from the emis-sion point with wind presence [20]. However, in its basic scheme, the gaussian model does not consider the molecular mass of the particles. Despite this limitation, it is still among the most helpful dispersion models, with various improvements and modifications published. Among others, a notable work was done by [21], in which a filament dispersion model was simulated in ROS alongside simulated gas sensors to manipulate various parameters and their combina-tions. However, not much has been published regarding gas dispersion models that simulta-neously consider the environmental structure and its conditions, except GADEN. Various surveys have acknowledged it as the best indoor gas dispersion model to study multiple gas dispersion scenarios through simulations. GADEN is an open-source simulator implemented in ROS as a ROS package based on the filament dispersion model [22]. GADEN comes with the functionalities to import 3D CAD models, simulate different gas sensors, and simulate multiple gas sources under certain indoor and wind conditions [2]. Hence, GADEN was found to be the ultimate choice for simulating and studying indoor gas dispersion scenarios.

## Use of reinforcement learning for path planning

This section briefly mentions recent studies that investigated or compared Reinforcement Learning (RL) with others to identify their effectiveness. A recent survey on gas source locali-zation techniques declared Deep RL the best among state-of-the-art in making the robots localize odors [22]. Model-free agents are usually selected where the nature of the environment is highly dynamic and tedious to model [23]. Another similar work endorsed that RL has proven to be the most promising technique to find flexible solutions and model-free RL can help develop effective strategies without prior knowledge [5]. The RL agent in action for this work with its background is presented ahead, under the subsections 'agent selection' and 'Q-learning.' Formalizing the problem as an RL problem is covered in detail under the section 'RL and gas concentration mapping.'

Some of the most cited and relatable literature has been presented in the form of a table, most cited first, comparing their strengths and this work which generates the path based on

learning methods. The viewpoints for evaluating the works cited below are based on a few critical points. First, an invited review paper on recent progress and trend in robot odor source localization [22], utilization of drones—since both gases and drones are airborne, and use of contact-based sensors which requires the robot and sensor to be in the gas dispersion environment. Table 1 summarizes the comparison between this and some most cited / recent work on a similar track, *i.e.*, gas source localization through robots.

## Work breakdown

The whole work was split into five sequential phases, depicted in Fig 1, with the nature of tasks based on knowledge areas described briefly. The thought behind designing an array of onboard sensors for the aerial robots to collect the data was conceived. Later, the acquired data would be fed to the machine learning algorithm, whose output would be expected to generate the robot's path steps.

In parallel to the division of work phases, multiple objectives were identified and grouped according to the literature findings under the flow chart of activities (Fig 2). This flow chart also highlights the flow of this article, and sections covered ahead can be seen under these sets of activities.

## Data acquisition

This section presents the data acquisition phase through two aspects: 1). Collection of gas dispersion data in an indoor (gymnasium hall) environment where temperature and humidity are known. At the same time, the wind speed was controlled by controlling the speeds of pre-installed exhaust fans in the gymnasium. 2). The drone path was generated in simulations through the RL agent, where the generated path was validated from the experimentation data. It was required to generate and collect gas dispersion data without the direct influence of outdoor wind and light conditions to be used as training and testing data.

Secondly, the sensors were supposed to collect the data while moving around, as mounted at UAV. However, it was impossible to emit the gas from the source continuously and deliberately involve means of obstructions in the natural flow of gas plume movement.

## Workspace planning

The whole vicinity was mapped as a 28 by 17 meters grid space with each cell size 1 by 1 meter. It was planned to disperse the gas from different distances and directions. So that the data collected through this would be used to train the AI for correct source location estimation; however, it was impossible to release gas for longer durations; consequently, the scenario was repealed, as seen in Fig 3. It was chosen to disperse gas for shorter durations at multiple locations, placing the sensory module in the center and collecting the data.

## Taguchi's orthogonal array for experiment design

Since the source of gas dispersion was moved at different angles and distances, it was ensured that the gas dispersion experiments covered the combinations mentioned in Table 2. There could have been a fair chance to omit an important location to conduct gas dispersion and include a lesser significant one. Taguchi's method was studied and based on this approach to address this concern, and the sensor readings were considered independent variables. Each sensor's possible readings were classified as high, medium, or low levels of gas readings. Based on this, Taguchi's orthogonal array with 4 parameters (P) and 3 levels (L) was considered (also known as P4L3), and the guidelines provided by [34] were followed.

**Table 1. A brief comparison of the presented work with the most cited and recent works on the similar topic.**

| Year | Title | Cited Work | Ref. | Presented Work |
|---|---|---|---|---|
| 2019 | Smelling Nano Aerial Vehicle for Gas Source Localization and Mapping | The cited work effectively utilized a very small UAV that collects gas readings during its flight in an indoor environment. A gas map is established from dozens of readings collected by the onboard gas sensors. The sensor is placed precisely at the top of the UAV. | [24] | Instead of one, four sensors are placed at distances with minimum impact of the air thrust from propellors. The readings of four sensors serve as input to the reinforcement learning agent that immediately generates a path toward the source without traversing through the plume. |
| 2011 | Collective Odor Source Estimation and Search in Time-Variant Airflow Environments Using Mobile Robots | The cited work engages multiple ground robots for odor source localization. Each robot has an anemometer and collision-avoiding sensors to efficiently navigate without colliding. | [25] | The presented work believes airborne robots have a natural advantage when dealing with gas source localization. It is a pricey limitation for gas source localizations when the robots are restricted to detect just a couple of feet above the ground. |
| 2013 | Optimal spatial formation of swarm robotic gas sensors in odor plume finding | Multiple ground robots were used to validate an optimized place of fixed-position gas sensors. | [26] | Secondly, this work utilizes the difference in time of arrival of gas sensor readings at all four sensors to estimate wind vectors without an anemometer. |
| 2011 | Olfaction and Hearing-Based Mobile Robot Navigation for Odor/Sound Source Search | The cited work uses two types of robots equipped with gas sensors and microphones to navigate sound and odors. The robots move toward the location of multiple gas sensors placed in the plume and generate a path. | [27] | The path shapes from the cited work and the work presented here significantly differ in the number of turns and directions away from the source. Secondly, as stated above, the advantage of this work is to utilize UAVs. |
| 2019 | Pollution Source Localization Based on Multi-UAV Cooperative Communication | The cited work engages three UAVs that use the Artificial Potential field and Particle Swarm Optimization, where each UAV acts as a particle. However, if any UAV detects a pollution source, the process completes. | [28] | The presented work generates a path to the emission point from detection. However, the cited work assumes the detection as localization through multiple UAVs. |
| 2019 | Chemical Source Searching by Controlling a Wheeled Mobile Robot to Follow an Online Planned Route in Outdoor Field Environments | The cited work uses a ground robot equipped with a chemical fume-sensing sensor and establishes a path from a continuous emission source considering the airflow of the region. The robot's speed is kept slow because of the slow response from the sensors. | [29] | The presented work utilizes the RL agent's training experience, enabling the agent to generate its path as soon as the four onboard sensors collect gas readings. Hence, this helps in avoiding any unnecessary exploration to reach the target. |
| 2020 | 3D Odor Source Localization using a Micro Aerial Vehicle: System Design and Performance Evaluation | The cited work used "crazy fly 2," named micro aerial vehicle, to detect odor with the sensor placed precisely on top and later moved to the bottom. CF2 is affected by wind speeds higher than 1m/sec. However, the prime focus was on the analysis of sensor placement. | [30] | The presented work is based on a study [10] that recommends the best distance to place gas sensors from the propellers, 1 to 1.2 meters from the propellors. |
| 2021 | Real-time odor concentration and direction recognition for efficient odor source localization using a small bio-hybrid drone | The cited study presented the development of an onboard odor sensor to estimate direction and source using a bio-hybrid drone. The sensor must be enclosed in a tube-like structure to avoid false direction detection. | [31] | The presented work used a similar idea to estimate the source location from the sensor reading's strength; however, in this case, the sensors were kept open. The source distance and direction were based on the readings of all four sensors, adding redundancy in the sensor's readings to make them more reliable. |
| 2019 | Wind-Independent Estimation of Gas Source Distance from Transient Features of Metal Oxide Sensor Signals | The cited work focuses on the data processing of the signals received by the gas sensors to overcome the intermittent detections in the form of patches, presenting the detection as an estimation problem. The work recommends using machine learning techniques in their future work. | [32] | The presented work already uses model-free reinforcement learning agents to estimate the plume source. |
| 2020 | Towards Fast Plume Source Estimation with a Mobile Robot | The experiment uses a ground robot in a wind tunnel-like environment with an air speed of up to 1m/sec, using three onboard MOX sensors as an odor compass. The plume is estimated through a particle-filter model, while the path is established through a Genetic Algorithm based approach. | [33] | The presented work belongs to the learning methods for gas source estimation, while the cited work used bioinspired. The present work considers a UAV, while the cited work uses UGV. Conducting experiments under a wind tunnel may offer more accuracy; however, the wind-tunnel scenarios take the experimentations a bit farther from realistic environments. |

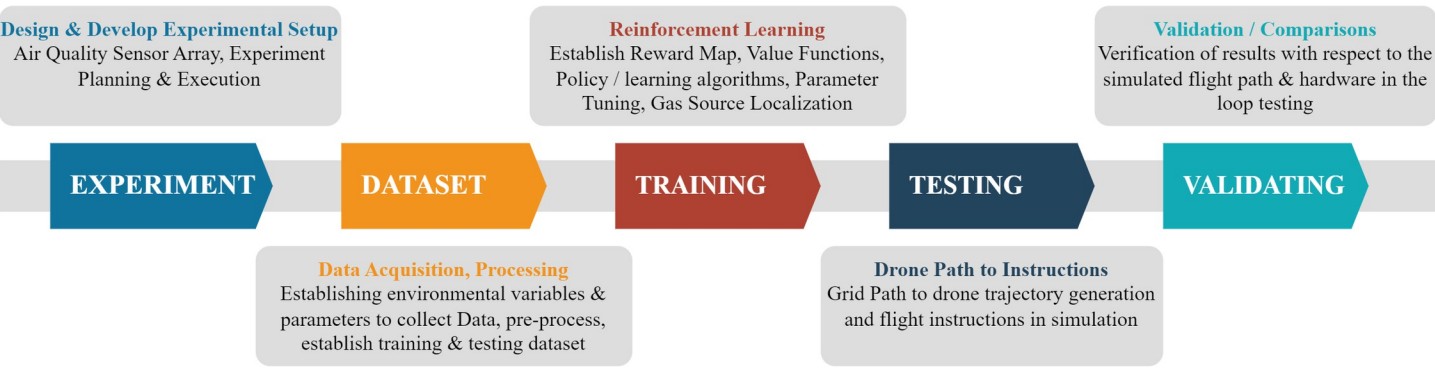

**Fig 1. Division of work phases.**

## Gas dispersion points for data collection

The indoor gas dispersion workspace to disperse gas and collect plume data comprised 28 by 17 meters, as mentioned already, with a wind speed of about 3 meters per second. The experiment workspace was well labeled with distance and direction markers at every half a meter and angle of 45˚, respectively. An illustration showing the experiment workspace has been presented in Fig 4, where the red dots represent the gas source points, and the sensors array is placed at the center of the environment. S0-S3 denotes sensors, and *P0-P3* are the positions of

**Fig 2. Activities flow chart.**

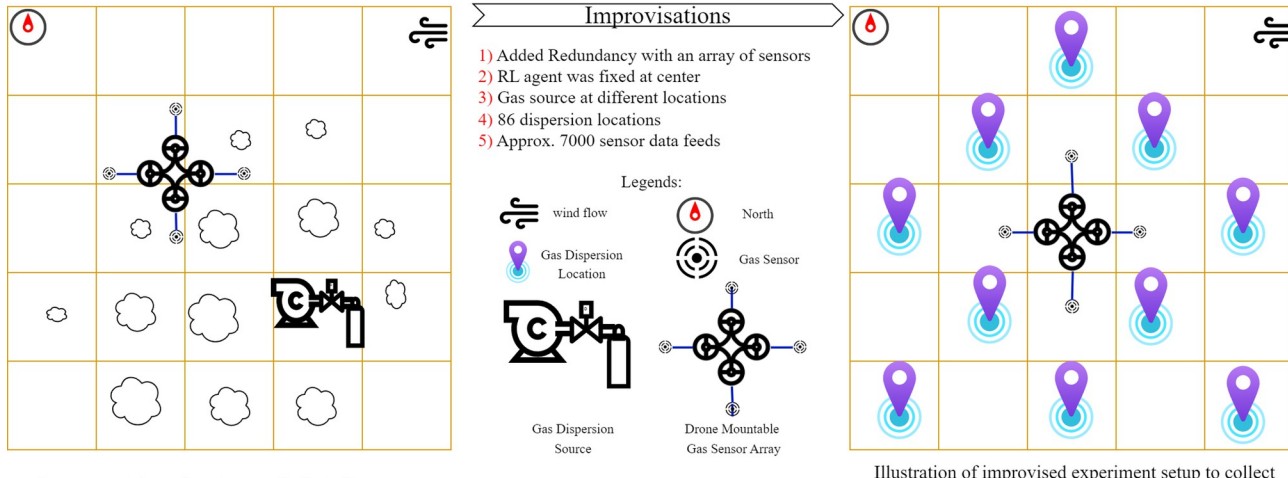

**Fig 3. Improvisations to collect data for an onboard-moveable array of sensors.**

propellers. Although Taguchi's method reduced the number of iterations to just a minimum of nine, these combinations could have been achieved by just covering one segment of the experimental workspace. Therefore, it was decided to cover all surrounding areas and ensure that the sensor data included the combinations in Table 2.

## Air Quality Sensors Array (AQSA)

### Sensor selection and specifications

There are various categories of gas sensors depending upon the mechanisms and methods for detecting gas molecules. However, the gas detection sensors are primarily chosen based on their application, accuracy, cost, availability, and reliability; a few other categories to mention are:

i. Electrochemical sensors: These sensors use a chemical reaction to detect the presence of gases and are commonly used to detect toxic gases such as carbon monoxide and hydrogen sulfide.

ii. Infrared sensors: These sensors use infrared technology to detect gases by measuring the absorption of infrared light by the gas molecules. They are commonly used to detect gases such as methane and carbon dioxide.

**Table 2. Taguchi's orthogonal array for 4 parameters 3 levels (P4L3).**

| Exp.# | s0 | s1 | s2 | s3 |
|---|---|---|---|---|
| 1 | Low | Low | Low | Low |
| 2 | Low | Medium | Medium | Medium |
| 3 | Low | High | High | High |
| 4 | Medium | Low | Medium | High |
| 5 | Medium | Medium | High | Low |
| 6 | Medium | High | Low | Medium |
| 7 | High | Low | High | Medium |
| 8 | High | Medium | Low | High |
| 9 | High | High | Medium | Low |

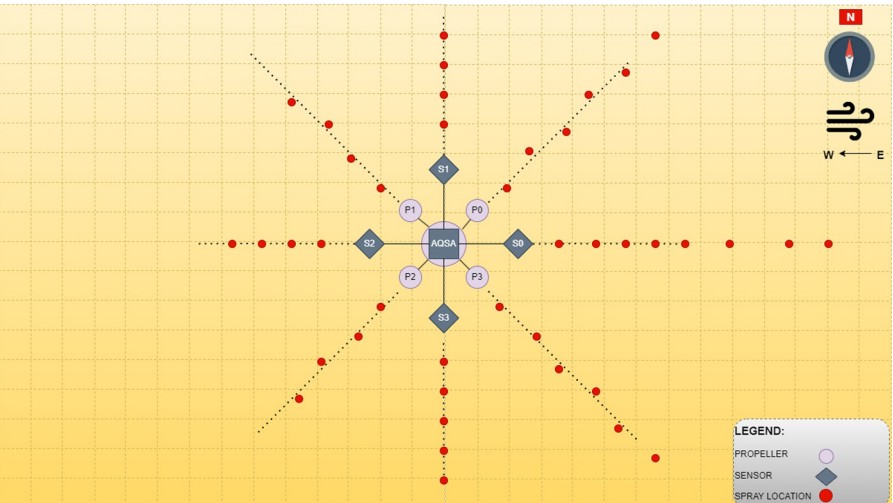

**Fig 4. Illustration of gas dispersion points at multiple angles and distances from the array of sensors.**

iii. Metal oxide semiconductor (MOS / MOX) sensors: These sensors use a MOS transistor to detect gases by measuring changes in conductivity. They are commonly used to detect gases such as carbon monoxide and hydrogen.

iv. Catalytic sensors: These sensors use a catalytic element to detect gases by measuring changes in the element's resistance due to the gas's presence. They are commonly used to detect gases such as propane and natural gas.

v. Ultrasonic sensors: These sensors use ultrasonic waves to detect the presence of gases. They are commonly used to detect gases such as methane and propane.

vi. Laser sensors: These sensors use laser beams to detect gases by measuring the absorption or scattering of the laser light by the gas molecules. They are commonly used to detect gases such as ammonia and chlorine.

## Metal Oxide (MOS / MOX) sensors

MOX sensors are commonly used to detect gases such as carbon monoxide, methane, and propane. They are relatively inexpensive and widely used in consumer, industrial, and automotive applications. They are also used in portable gas detectors, which makes them accessible and inexpensive. The sensor utilized in this particular study is MQ5 whose data sheet and temperature sensitivity is presented under reference [35].

**Advantages & limitations of MOX sensors.** Following are a few advantages associated with MOX sensors:

I. Low cost of production

II. High sensitivity

III. Fast response time

IV. Low power consumption

V. Readily available from the local market

VI. Widely used for indoor gas detection

The limitations of MOX sensors include:

i. They have limited selectivity, meaning they can detect multiple gases.

ii. They can be affected by humidity and temperature changes.

iii. They have a limited lifespan and need to be replaced periodically.

## Strategies incorporated to overcome the limitations

The first limitation is not faced because of using just one gas for experimentation.

For the second limitation, an investigation on the effects of environmental temperature and humidity levels on MOX gas sensor response was considered which indicated that the variations in temperature were found to cause a drift the base reading of the gas sensor. Most of the sensors which were tested in the study demonstrated decreasing outputs over a rise in temperature, however an increase was exhibited over an opposite pattern [36]. This led to deduce that the drift of responses from the MOX gas sensors are needed to be taken into consideration in an environment with temperature variabilities. However, this limitation remains ineffective because of the near-constant weather of Kuala Lumpur where the gas dispersion experiments were conducted in an indoor environment.

**Factor of pressurized gas container.** The gas used for the dispersion experiment was available in pressurized gas containers (butane cartridge weighing 280 grams) and when a compressed gas is released from a pressure vessel, it often undergoes 'adiabatic expansion'. Adiabatic processes involve the release or absorption of heat without the transfer of heat to or from the surroundings. During rapid expansion, the gas can cool or heat up, leading to a temperature change. Heat transfer can also occur between the gas and the walls of the pressure vessel or surrounding environment. The temperature of the gas can be influenced by the initial temperature of the pressure vessel, as well as ambient conditions [36].

The effect of this particular factor is not considered in this study and has mentioned under the limitations of this work. However, there are a few reasons to believe that the effect of this factor would be least on the results of this work:

i. The location of source does not directly rely on the gas readings, in fact, the release of gas at different distances and angles from a gas plume and the role of gas sensors is to identify the concentration level at different zones of a gas plume rather than identifying the source.

ii. The collected data is used to train machine learning algorithms by supervising the learning agent with the correct place of the gas source. This eventually trains the AI (RL agent) to detect the source from this data. Thus, the AI trained to detect the source location includes this anomaly within the training data.

iii. During the gas dispersion experiment for data collection, the distance between the gas distribution points and the senor array ranged from one up to ten meters as shown in Fig 4. It can be inferred that by the time the gas molecules would have reached the sensors array, the temperature and pressure of the indoor environment have reduced this factor significantly.

Two other critical limitations were considered and addressed in this work while dealing with the MOX sensors for gas detection.

**The lifespan of MOX sensors, when to change the MOX sensor?.** A high-quality hand-held sensor was utilized to calibrate the reading of all sensors involved in the gas dispersion data collection, upon poor reading, the sensors were replaced.

**The random nature of gas particles.** Due to the random nature of gas particles, the sensors may generate inconsistent readings; however, the problem was taken care of by introducing an array of sensors rather than relying on just one. The Air Quality Sensor Array (AQSA) design contains four sensors at specific angles and distances from each other which adds redundancy.

**Pre-heating of MOX sensors.** MOX sensors require 24 hours of pre-heating, based on the data sheet from the supplier, and any difference in pre-heating while using more than one sensor in an experiment can question the accuracy of the readings. Therefore, in case of one malfunction sensor, all four sensors from the array were replaced to tackle this issue.

## Sensor's placement

As stated earlier in Table 1, the inspiration behind the design was published research in which the impact of air thrust from the rotary blades of UAV was investigated [10], and the most suitable distance of the gas sensor from the position of the UAV's propeller thrust, 1 to 1.2 meters. Off-the-shelf Metal Oxide (MOX) sensors were selected to realize the idea of a sensor array, as illustrated in Fig 5. With all the distances known between any two sensors, the time difference of arrival of change in the sensor's reading can help to estimate the wind vector. Moreover, the differences in the magnitude of two adjacent sensors can help identify the direction of the gas plume's emission source. The proposed design is a novel approach and one of the significant contributions of this work with the following features to mention:

a. The placement of MOX sensors ensured to bear the minimum impact of air thrusts from the UAV's propellors.

b. The chances of disconnected patches are decreased due to a larger span of the sensors array.

c. Identification of wind speed and direction without using any anemometer.

d. Multiple sensors onboard ease the impact of the random nature of gas particles.

e. The added redundancy also helps detect any sensor malfunctioning.

## Data pre-processing

The data were normalized as the sensors may have different starting values and pre-processed to find the overall minimum, maximum, mean, and first-order derivatives for every sensor in each experiment. The experiment workspace was considered a grid space, and data view comprising the timestamp, spray location, angle from source, and distance from the source, and sensor readings were collected, as shown in Fig 6. The data will train reinforcement learning agents to estimate the distance and direction from the emission point.

## Trajectory mode selection

It is essential to mention that the activity is planned to be executed in three phases: Search, Track, and Localize gas emissions. This work covers track and localization out of these three trajectory modes, while the search mode is beyond scope. These modes depend upon the readings from gas sensors and are presented in Fig 7. If there are no readings, or the highest

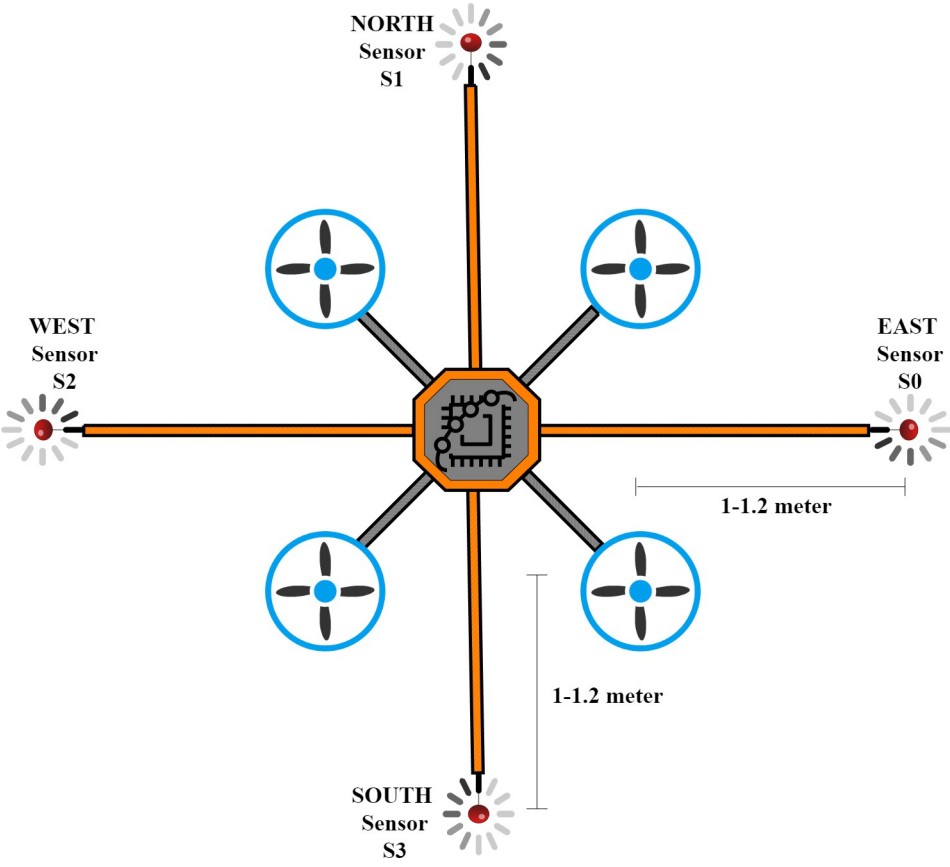

**Fig 5. Illustration of sensors placement.**

reading from any sensor is below 0.09 parts per million (ppm), the agent must stay in search mode and keep looking for gas traces. The value 0.09 ppm has been taken from the experiment data to avoid sensor inaccuracies or unreliable detections of small patches. In this mode, the

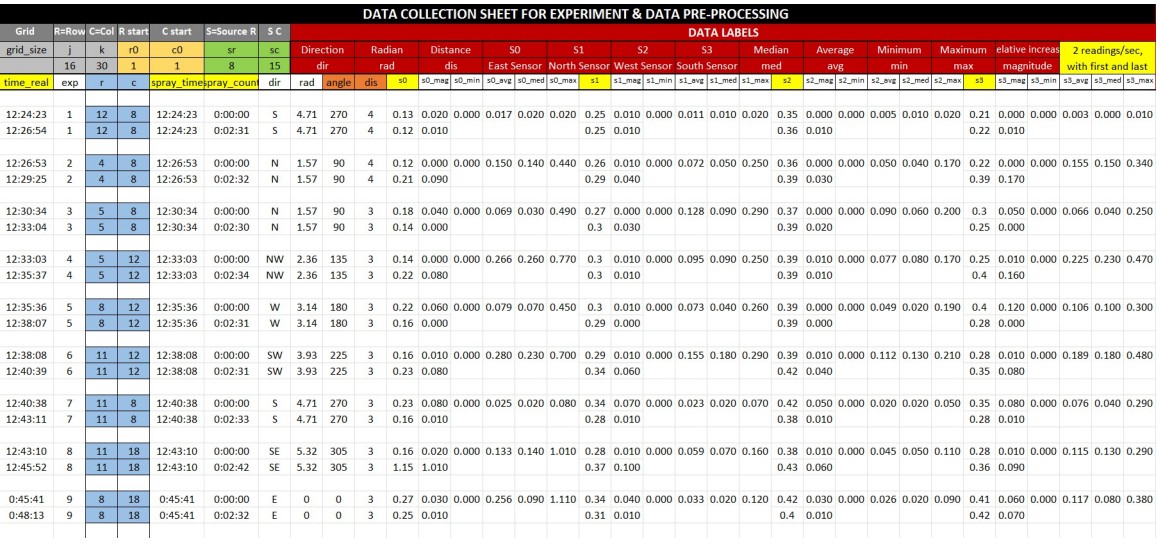

**Fig 6. Data collection sheet showing the set of variables.**

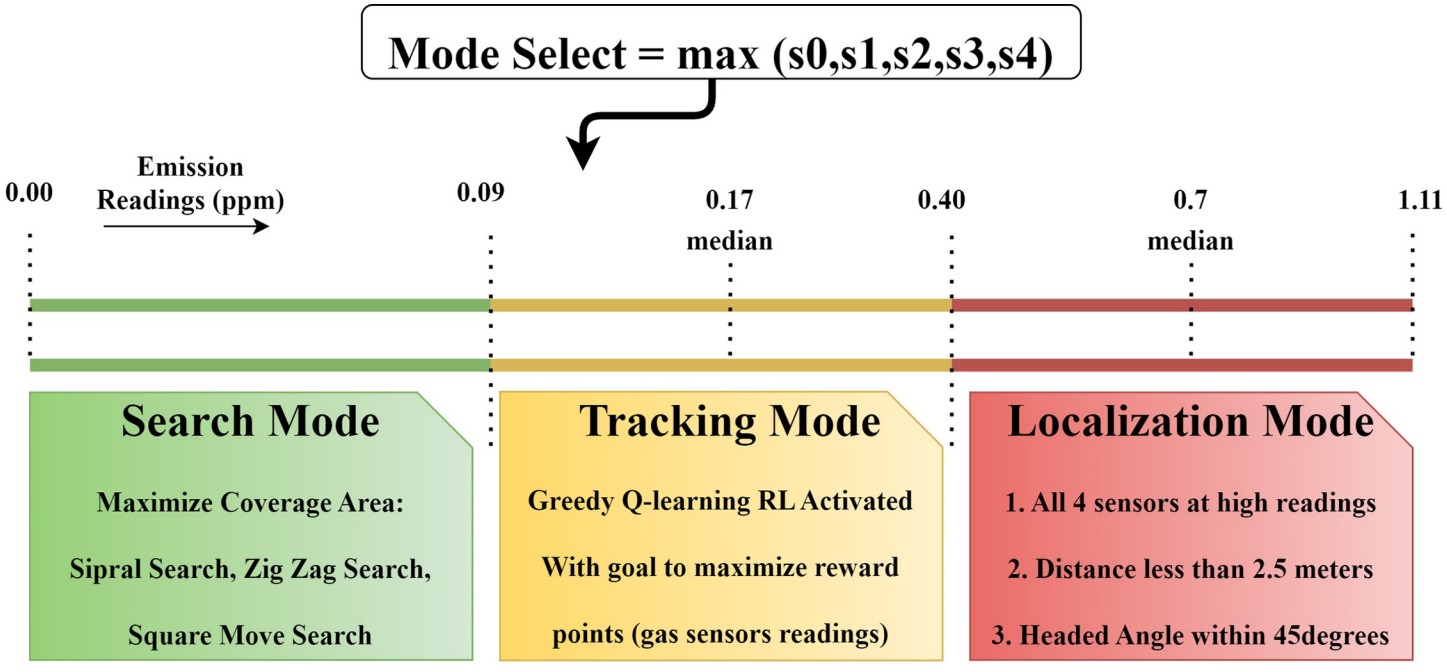

**Fig 7. Trajectory mode selection.**

UAV would follow a pre-programmed area coverage path, which be a square move, spiral search, zigzag, or raster scan pattern, which are not covered in the scope of this work. As soon as any of the gas sensor reading crosses the set threshold, the UAV would shift to tracking mode. The reinforcement learning agent would generate the path immediately regarding direction and the number of steps to take. However, the scope of this work is limited to the autonomous path generation to the plume source.

## Gas concentration mapping and source estimation

### Reinforcement learning and gas concentration mapping

Before describing the reinforcement learning (RL) technique designed for this work, it is vital to state the mathematics behind the nature of the problem that an RL agent tries to solve, the Markov Decision Process (MDP). A finite MDP is a mathematical framework used to model decision-making situations where a decision depends on the system's current state and the system's future conditions are uncertain. Additionally, a reward function is defined that assigns a real-valued reward to each state-action pair. The goal of an MDP is to find an optimal policy, which is a mapping from states to actions that maximize the expected cumulative reward over time [37].

The basis for formulating this problem has been laid through the finite Markov Decision Process (MDP) in which an agent interacts with its environment to reach its goal. In MDPs, the three fundamental aspects-sensation, goal, and actions-are needed to consider any problem to be solved through an RL method [38]. The whole paradigm of reinforcement learning comprises the following components:

- Agent: The entity that interacts with the environment and takes action.

- Environment: The system that the agent interacts with to change the state.

- State: The current situation or configuration of the environment.

- Action: The choices available to the agent in each state.

- Reward: A scalar value indicates the agent's performance in the environment.

- Policy: A function that maps states to actions.

- Value function: A function that estimates the long-term value of a state or an action.

- Model of the environment: A simplified representation of the environment that the agent can use to predict the outcome of its actions.

  Fig 8 illustrates the agent-environment interaction and the flow of data processing

- The environment in this work is a 2D grid map with each cell accessible through a row and column value.

- The agent is responsible for the drone path generation and has a set of four actions: up, down, left, and right to move around in the environment.

- The positive and negative rewards are given to the agent on each move based on the reward map.

- The highest reading from any of the four sensors is considered the 'max' reading and is deemed zero distance from the source because of the sensor's limitation.

- The negative of the 'max' is treated as the boundary of the environment.

- An arithmetic progression is used between two consecutive gas sensor readings to fill in the values from the boundaries to the source.

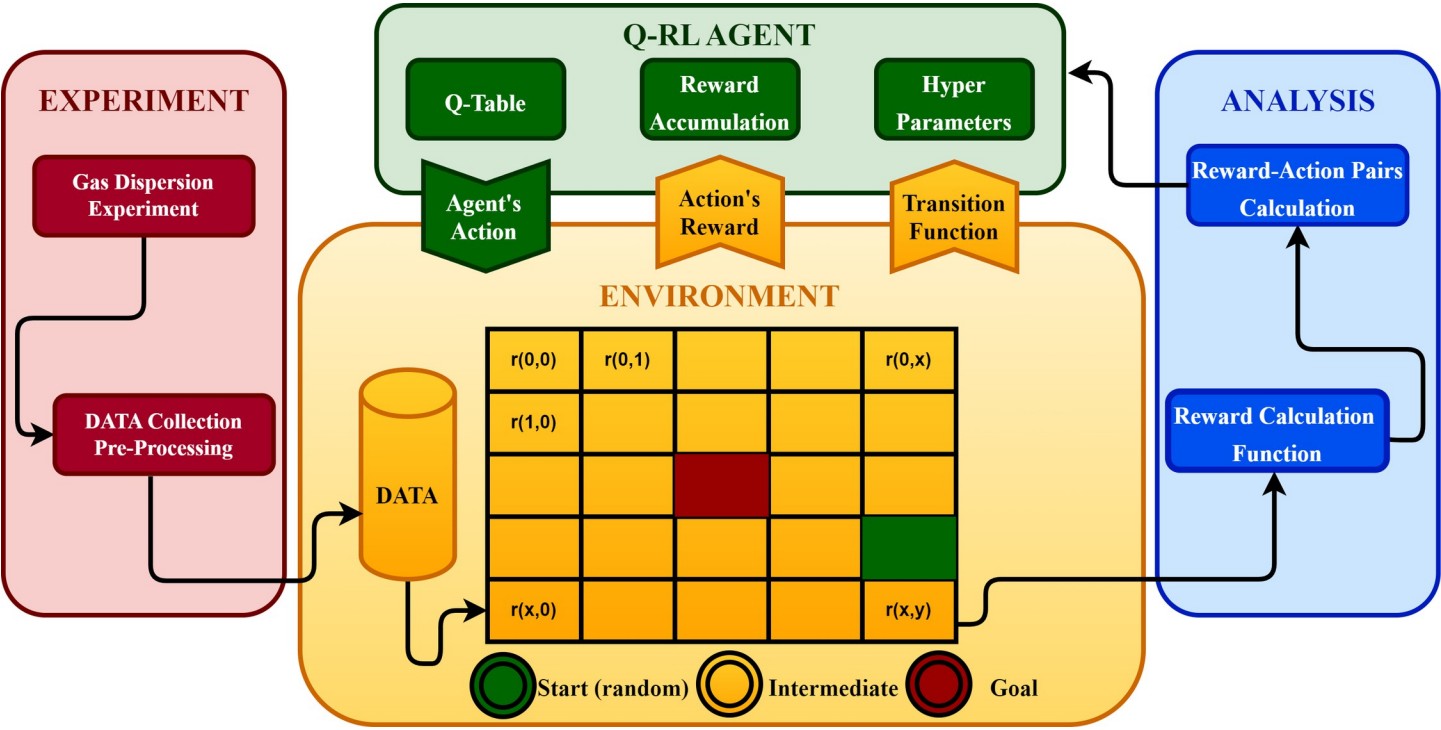

**Fig 8. Agent-Environment interaction.**

- The objective is to encourage the agent to accumulate and maximize rewards (higher gas concentrations).

## Reward formulation

The MOX sensors need to be pre-heated to get in function. The pre-heating duration while dealing with multiple sensors for different iterations of experiments can produce a base difference in sensor readings. For this reason, only the amount of change in a sensor's reading is considered. Table 3 presents the reward formulation with the help of the sensor's data and highlights the maximum, minimum, and median reward for each trajectory mode. The table legends are also mentioned with it.

## Reward map generation

Since the rewards are formulated based on Table 2, the agent's environment is populated with reward points to serve as a navigation map based on varying gas concentration levels. The established reward map to train the RL agent for its navigation toward the source of emission is presented in Fig 9.

**Table 3. Reward Formulation.**

| Sensor Data Significant Observations | | | |
|---|---|---|---|
| Highest Sensor Reading (ppm) | Trajectory Modes | Reward | Value |
| | | maximum | 1.11 |
| > 0.4 | Localization Mode | Localization: Median | 0.7 |
| 0.09>AND < 0.4 | Tracking Mode | Tracking: Median | 0.17 |
| ELSE | Searching Mode | Minimum | 0.09 |
| | | Boundary: -1*(Max) | -1.11 |
| | | Overall Median | 0.445 |
| Reward Formulation for Step Size | | | |
| Horizontal Step Size (Wind Favor), h1 | | | 0.19 |
| Horizontal Step Size (Wind Against), h2 | | | 0.145 |
| Vertical Step Size (Up/Down/Diagonal),v | | | 0.28 |

The terms used in reward formulation for each grid cell:

a_max = The maximum reading from all sensors

Source Location = Location of a_max

Grid Boundary = -1 * (a_max)

Horizontal Step Size1 = h1

Horizontal Step Size2 = h2

Vertical Step Size = v

Reward Step (r) calculation moving from source point towards boundary:

UP: next_cell_r = previous_cell_r–v

DOWN: next_cell_r = previous_cell_r–v

LEFT: next_cell_r = previous_cell_r–h2

RIGHT: next_cell_r = previous_cell_r–h1

| Reward Points Calculations (Drone Navigation Map - ideal Distribution of Base Reward Values with effectiveness of wind) | | | | | | | | | | | | | | | | | | | | | | | | | | | | |
|---|---|---|---|---|---|---|---|---|---|---|---|---|---|---|---|---|---|---|---|---|---|---|---|---|---|---|---|---|
| r\|c | | 1 | 2 | 3 | 4 | 5 | 6 | 7 | 8 | 9 | 10 | 11 | 12 | 13 | 14 | 15 | 16 | 17 | 18 | 19 | 20 | 21 | 22 | 23 | 24 | 25 | 26 | |
| | -1 | -1.1 | -1.1 | -1.1 | -1.1 | -1.1 | -1.1 | -1.1 | -1.1 | -1.1 | -1.1 | -1.1 | -1.1 | -1.1 | -1.1 | -1.1 | -1.1 | -1.1 | -1.1 | -1.1 | -1.1 | -1.1 | -1.1 | -1.1 | -1.1 | -1.1 | -1.1 | -1 |
| 1 | -1 | -1.1 | -1.1 | -1 | -1 | -1 | -0.9 | -0.9 | -0.9 | -1.1 | -1.1 | -1.1 | -1.1 | -1.1 | -1.1 | -0.9 | -1.1 | -1.1 | -1.1 | -1.1 | -1.1 | -1.1 | -0.9 | -0.9 | -1 | -1 | -1.1 | -1 |
| 2 | -1 | -1.1 | -1 | -1 | -0.9 | -0.8 | -0.8 | -0.7 | -0.6 | -0.6 | -0.7 | -0.7 | -0.8 | -0.8 | -0.8 | -0.6 | -0.8 | -0.8 | -0.8 | -0.7 | -0.7 | -0.6 | -0.7 | -0.8 | -0.9 | -1 | -1.1 | -1 |
| 3 | -1 | -1.1 | -1 | -0.9 | -0.8 | -0.7 | -0.7 | -0.6 | -0.5 | -0.4 | -0.3 | -0.4 | -0.4 | -0.4 | -0.5 | -0.3 | -0.5 | -0.4 | -0.4 | -0.4 | -0.3 | -0.4 | -0.6 | -0.7 | -0.8 | -1 | -1.1 | -1 |
| 4 | -1 | -1.1 | -1 | -0.9 | -0.8 | -0.7 | -0.6 | -0.5 | -0.3 | -0.2 | -0.1 | -0 | -0.1 | -0.1 | -0.1 | -0 | -0.1 | -0.1 | -0.1 | -0 | -0.2 | -0.3 | -0.5 | -0.6 | -0.8 | -1 | -1.1 | -1 |
| 5 | -1 | -1.1 | -1 | -0.9 | -0.7 | -0.6 | -0.5 | -0.4 | -0.2 | -0.1 | 0 | 0.1 | 0.3 | 0.2 | 0.2 | 0.3 | 0.2 | 0.2 | 0.3 | 0.1 | -0.1 | -0.2 | -0.4 | -0.6 | -0.8 | -0.9 | -1.1 | -1 |
| 6 | -1 | -1.1 | -1 | -0.8 | -0.7 | -0.6 | -0.4 | -0.3 | -0.1 | -0 | 0.1 | 0.3 | 0.4 | 0.6 | 0.5 | 0.6 | 0.5 | 0.6 | 0.4 | 0.2 | -0 | -0.2 | -0.4 | -0.6 | -0.7 | -0.9 | -1.1 | -1 |
| 7 | -1 | -1.1 | -1 | -0.8 | -0.7 | -0.5 | -0.4 | -0.2 | -0.1 | 0.1 | 0.2 | 0.4 | 0.5 | 0.7 | 0.8 | 0.8 | 0.8 | 0.6 | 0.4 | 0.2 | 0.1 | -0.1 | -0.3 | -0.5 | -0.7 | -0.9 | -1.1 | -1 |
| 8 | -1 | -1 | -0.8 | -0.7 | -0.5 | -0.4 | -0.2 | -0.1 | 0.1 | 0.2 | 0.3 | 0.5 | 0.6 | 0.8 | 0.9 | 1.1 | 1 | 0.8 | 0.6 | 0.4 | 0.2 | 0 | -0.2 | -0.4 | -0.5 | -0.7 | -0.9 | -1 |
| 9 | -1 | -1.1 | -1 | -0.8 | -0.7 | -0.5 | -0.4 | -0.2 | -0.1 | 0.1 | 0.2 | 0.4 | 0.5 | 0.7 | 0.8 | 0.8 | 0.8 | 0.6 | 0.4 | 0.2 | 0.1 | -0.1 | -0.3 | -0.5 | -0.7 | -0.9 | -1.1 | -1 |
| 10 | -1 | -1 | -1 | -0.8 | -0.7 | -0.6 | -0.4 | -0.3 | -0.1 | -0 | 0.1 | 0.3 | 0.4 | 0.6 | 0.5 | 0.6 | 0.5 | 0.6 | 0.4 | 0.2 | -0 | -0.2 | -0.4 | -0.6 | -0.7 | -0.9 | -1.1 | -1 |
| 11 | -1 | -1 | -1 | -0.9 | -0.7 | -0.6 | -0.5 | -0.4 | -0.2 | -0.1 | 0 | 0.1 | 0.3 | 0.2 | 0.2 | 0.3 | 0.2 | 0.2 | 0.3 | 0.1 | -0.1 | -0.2 | -0.4 | -0.6 | -0.8 | -0.9 | -1.1 | -1 |
| 12 | -1 | -1.1 | -1 | -0.9 | -0.8 | -0.7 | -0.6 | -0.5 | -0.3 | -0.2 | -0.1 | -0 | -0.1 | -0.1 | -0.1 | -0 | -0.1 | -0.1 | -0.1 | -0 | -0.2 | -0.3 | -0.5 | -0.6 | -0.8 | -1 | -1.1 | -1 |
| 13 | -1 | -1.1 | -1 | -0.9 | -0.8 | -0.7 | -0.7 | -0.6 | -0.5 | -0.4 | -0.3 | -0.4 | -0.4 | -0.4 | -0.5 | -0.3 | -0.5 | -0.4 | -0.4 | -0.4 | -0.3 | -0.4 | -0.6 | -0.7 | -0.8 | -1 | -1.1 | -1 |
| 14 | -1 | -1.1 | -1 | -1 | -0.9 | -0.8 | -0.8 | -0.7 | -0.6 | -0.6 | -0.7 | -0.7 | -0.8 | -0.8 | -0.8 | -0.6 | -0.8 | -0.8 | -0.8 | -0.7 | -0.7 | -0.6 | -0.7 | -0.8 | -0.9 | -1 | -1.1 | -1 |
| 15 | -1 | -1.1 | -1.1 | -1 | -1 | -1 | -0.9 | -0.9 | -0.9 | -1.1 | -1.1 | -1.1 | -1.1 | -1.1 | -1.1 | -0.9 | -1.1 | -1.1 | -1.1 | -1.1 | -1.1 | -1.1 | -0.9 | -0.9 | -1 | -1 | -1.1 | -1 |
| | -1 | -1.1 | -1.1 | -1.1 | -1.1 | -1.1 | -1.1 | -1.1 | -1.1 | -1.1 | -1.1 | -1.1 | -1.1 | -1.1 | -1.1 | -1.1 | -1.1 | -1.1 | -1.1 | -1.1 | -1.1 | -1.1 | -1.1 | -1.1 | -1.1 | -1.1 | -1.1 | -1 |

**Fig 9. Reward distribution map for the training of RL agent (green to red: Higher to lower reward points).**

## Autonomous path generation

### Agent's policy selection

An agent's objective is to maximize its rewards by taking optimal actions at each state, and it is an agent's policy that decides these actions. The role of policy selection is vital depending upon the type of problem. Since the RL agent is required to identify the point of highest gas concentration in an environment that consists of gas particles with random nature, modeling such an environment to take-real time decisions sound impractical. Therefore, the agent's policy must be model-free, where the modeling information of the environment is not required.

Monte-Carlo and Temporal Difference (TD) are the two primary schools to set up model-free policies, and both have their uniqueness. When an agent goes through training, it may take thousands of iterations to learn about its environment, and one iteration is called an episode. The agents that develop their policies from the means of all observations during the training belong to Monte-Carlo-based learning policies, which are unsuited for the problem as consideration. As in gas source localization, any state an agent observes can potentially have exceptionally higher significance than the mean observations. However, in the temporal difference-based methods, the agent learns at state-level episodes, making these a more appropriate choice for the agent's policy [38].

Temporal difference offers Q-learning and SARSA (State-Action-Reward-State-Action), where Q refers to the 'Quality' of a decision. The difference between the two is on-policy and off-policy. SARSA is an on-policy scheme that sticks to the overall policy for choosing the following action, or it can compromise a higher immediate reward considering its long-term consequences. However, Q-learning is off-policy. Q-learning prefers a greedy action to look for the maximum Q-value for a state irrespective of long-term consequences. Ever since it has been believed that the gas concentrations (translated into rewards) would always be higher than before if the agent moves from any direction toward the point of emission. This clearly shows how Q-learning is a better-suited policy for the agent under consideration [38].

## Agent's priority selection

In Q-Learning, an exploration-exploitation strategy is offered and can be controlled through the help of hyperparameters. Exploring is done by taking random action and finding how good that action is for that state, and exploiting is done by taking that action with the maximum Q value.

The parameter to choose between explore and exploit epsilon, whose value is taken as 1 initially and decayed gradually as a function of episodes. The decaying can be linear, logarithmic, or exponential based on the requirement. A random number is generated, and if it turns out to be less than the epsilon value, a random action is taken. The greedy action is retained if the generated number exceeds the epsilon value. The results of varying hyperparameters are also collected and presented in the results section.

## Pseudo algorithm

The learning process can be improved by fine-tuning the three hyperparameters: epsilon, the discount factor, and the learning rate. Epsilon employs the agent to decide whether to explore the environment or exploit the existing Q-table. The discount factor influences the agent to pursue either immediate or future rewards, while the learning rate controls the speed or rate at which the agent takes to learn the environment.

The algorithm presented here takes the sensor readings, the total number of readings, and the hyperparameters to optimize the policy as input to generate the Q-Table for establishing State-Action pairs. The Bellman equation is utilized to determine how good an agent's current state can be, which uses the maximum expected rewards and discount rate with current rewards to update every entry in the Q-table. The Q-table is initialized with null entries, and as the agent starts exploring the environment, the Q-table gets populated. As the training progresses, this Q-value is updated and eventually converges to a higher positive value if the agent finds the optimal actions offering higher rewards. The process is presented in the form of a pseudo algorithm using the under-mentioned nomenclature:

$\varepsilon \rightarrow$ Epsilon
$\gamma \rightarrow$ discount factor
$\alpha \rightarrow$ learning rate
$N \rightarrow$ number of iterations
$s_k \rightarrow$ agent's state at $k^{th}$ sequence
$a_k \rightarrow$ agent's action at $k^{th}$ sequence
$Q_k \rightarrow$ Q value ($k^{th}$ entry in the Q-table)

**Pseudo algorithm: Sensors to Direction and Step Calculation of RL Agent**

```
Input: Learning parameters: Discount factor γ, Learning rate α,
    epsilon (exploration-exploitation trade-off ε, number of
    readings N, Sensor readings (s0, s1, s2, s3)

Input: Environment size (row, column)

1 Initialize Q₀ (s, a)← 0

2 for counter = 0:N do

  for k = 0,1,2... do

3    Generate a random number and compare it with epsilon ε

4    if random number > ε
```

**5** `Choose` $a_k$ `from` $Q_k$ `(`$s_k$`,` $a_k$`)`

**6** `Agent position = previous position + ak`

**7** **Else**

**8** **Calculate:** $angle = tan^{-1}\frac{\max(s0,s1,s2,s3)}{second\_\max\ (s0,s1,s2,s3)}$

**9** `Agent position = previous position + direction obtained from the angle`

**10** **Update:**

**11** $Q_{k+1}$ `(`$s_k$`,` $a_k$`)`$\leftarrow$ $Q_k$ `(`$s_k$`,` $a_k$`)`$+\alpha$`[`$r_k$ `+`$\gamma$`.max`$Q_{k+1}$ `(`$s_k$`,` `a)`$-$ $Q_k$ `(`$s_k$`,` $a_k$`)]`

**12** `Until counter = N(number of readings)`

## Generation of trajectories

The section primarily presents two key algorithms: the generation of the Q-table during the training process, where the agent keeps updating the table, and the generation of drone instructions with the help of the Q-table and sensor input during the testing process.

 **Updating the Q-table.** Fig 10 presents the algorithm for updating the Q-table iteratively. The Q-table is responsible for assigning Q-value to a candidate action in a state so that the

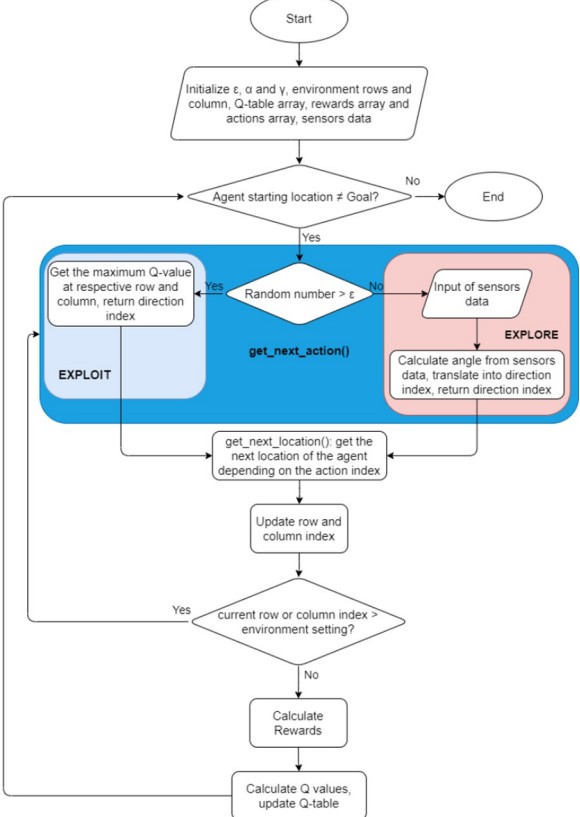

**Fig 10. Q-table generation and updating mechanism.**

agent has a quick-look table in state action pairs after the training. For each state-action pair, a q-value is allocated, and the q-value helps choose the best move at a specific state to reach the goal (maximum gas concentration reading) based on the reward map.

**Drone trajectory generation.** The next step is to generate a sequence of waypoints for the drones. These waypoints consider the sensor's input to determine a state with the help of a Q-table containing the state-action pairs against Q-values, selecting the best action (waypoint) for the drone to execute. Fig 11 presents the flowchart for the trajectory generation fed to the Hector quadrotor.

## Hyperparameters combinations and computational time

It is also essential to mention the impacts of hyperparameters. The epsilon variation, ε, affects the number of updates in the Q-table, γ, the discount rate that influences the agent to pursue immediate or future rewards, and α. This learning rate will affect the agent's learning rate. A factor of 0 α will make the agent learn nothing and keep exploiting the existing Q-table. In contrast, a factor of 1 makes the agent only use the current information—in this case, the sensor's values, and ignore the existing Q-table, just exploring the environment.

The parameter variation was divided into three states which were low, medium, and high. The low state was indicated by the value of 0.1, the medium state was indicated by the value of 0.5, and the high state was indicated by the value of 0.9. Fifteen combinations among these parameters were evaluated. The combination with the best performance would generate the nearest path to be fed in the UAV-Reinforcement Learning interface. The results of these combinations are presented in Table 4.

The first combination, ε = 0.9, γ = 0.9, and α = 0.9, produced the least overall computational time for the training with a period of 1258.8862 seconds. Combination number 1 also took the least time for computing the training at 1258.8802 seconds, as highlighted in light

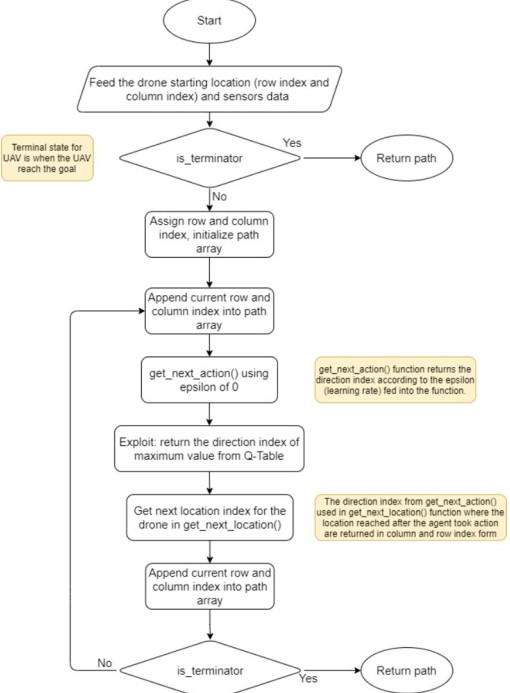

**Fig 11. Drone path calculation from the sensor input and Q-table.**

**Table 4. Hyper parameter optimization through trial and error.**

| Combination | Epsilon, ε | Discount factor, γ | Learning rate, α | Training Time (sec.) |
|---|---|---|---|---|
| 1 | 0.9 | 0.9 | 0.9 | 1258.8802 |
| 2 | 0.9 | 0.9 | 0.5 | 1307.7030 |
| 3 | 0.9 | 0.5 | 0.9 | 1413.3738 |
| 4 | 0.9 | 0.5 | 0.5 | 1412.2664 |
| 5 | 0.5 | 0.9 | 0.9 | 1409.5686 |
| 6 | 0.5 | 0.9 | 0.5 | 1461.7873 |
| 7 | 0.5 | 0.5 | 0.9 | 1372.4699 |
| 8 | 0.5 | 0.5 | 0.5 | 1412.0677 |
| 9 | 0.5 | 0.5 | 0.1 | 1400.6726 |
| 10 | 0.5 | 0.1 | 0.5 | 1398.3765 |
| 11 | 0.5 | 0.1 | 0.1 | 1390.9271 |
| 12 | 0.1 | 0.5 | 0.5 | 1386.0463 |
| 13 | 0.1 | 0.5 | 0.1 | 1400.2838 |
| 14 | 0.1 | 0.1 | 0.5 | 1420.2228 |
| 15 | 0.1 | 0.1 | 0.1 | 1414.4633 |

gold, while the slowest combination is combination number 6 at 1461.7873 seconds. The results are shown in Fig 12. The computational time for path generation, combination number 10, is the fastest at 0.0023 seconds, while the slowest is combination number three and seven at 0.009 seconds, respectively.

## Results and discussions

The split between testing and training datasets was 40% to 60% for validating the simulation results, respectively. 40% of the gas experiment data constituted thirty-one gas dispersion scenarios utilized for testing.

### Path-steps generation from random locations

The trajectory following the simulation for the UAV is generated using the best parameters variation combination, which is ε = 0.9, γ = 0.9 and α = 0.9 as observed under hyperparameter combinations. The RL agent was trained with this combination, and thirty-one trajectories were generated by giving random starting locations to the agent from the testing dataset. This means, with the help of its Q-table and the input from 4 sensors, the RL agent generated the sequence of trajectories to the emission point as presented in Table 5. The later subsection would also present the trajectories followed by Hector Quadrotor (a ROS-based UAV simulator) through the sequence of these steps to visualize the starting points, random source locations, and generated trajectories mapped over the environment's illustration.

The generated trajectory is validated as accurately as all the paths move toward the source location. This means the accuracy of this model validation is 100% in path generation; however, a few of these paths were unrealizable for the UAV simulated in ROS. Since the trajectories are not optimized for any specific robot and ROS considers the real-world limitations of robots, the problem of optimizing the path is beyond the scope here.

### Error computations for steps to trajectory conversion in ROS

The path generated from the RL agent, shown in Table 5, was fed to Hector Quadrotor UAV in ROS-Gazebo simulation to validate the generated path towards the gas source. In terms of

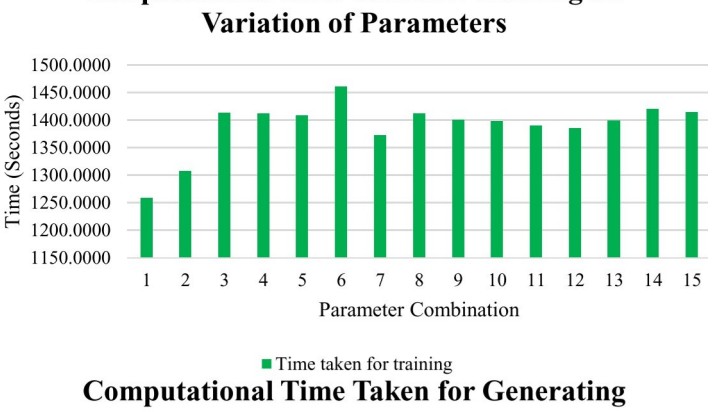

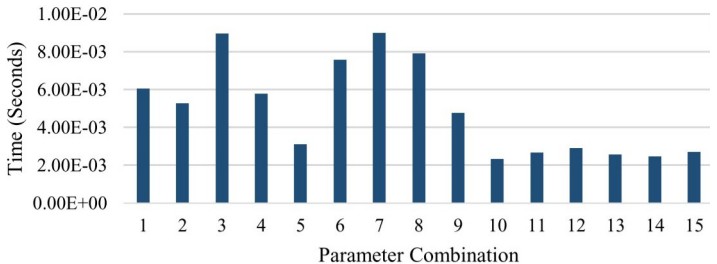

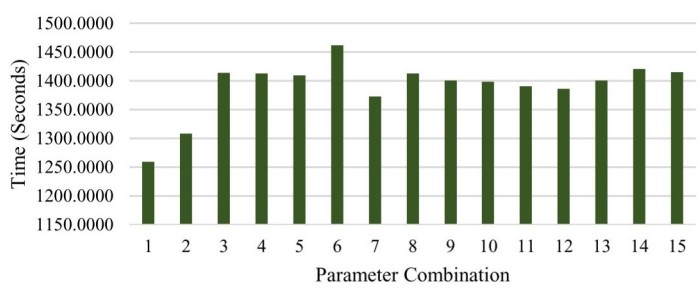

**Fig 12.** (a). Training Time vs. Combinations, (b). Path Generation time vs. Combinations, (c). Total Computational Time.

its ability, ROS is a simulation platform where the codes can be moved to the robot's hardware without any change. The algorithm's capability is evaluated by dividing the simulation results into two categories, reaching the source and heading to the source, as soon as all four sensors record hit 'high' reading as presented in mode selection. The reaching condition can only be satisfied when the distance between UAV and the source location is less than two and a half meters. There is no theoretical binding behind selecting this distance, except for the fact that the span of the drone through any axis is nearly three meters.

The heading of the source condition is satisfied when the path marked in the '**rqt_multi-plot**' from ROS in the next section shows the behavior of the UAV heading toward the source. Out of thirty-one scenarios, fifteen scenarios satisfied the reached the source condition while

**Table 5. Path generation from 31 unknown starting locations to source (15,8).**

| Scenario | Starting Column | Starting Row | Path-Steps Generated from 31 random locations ([row, col]) to Source at Column:15, Row:8 ([8,15]) |
|---|---|---|---|
| 1 | 16 | 7 | [[7, 16], [8, 15]] |
| 2 | 6 | 10 | [[10, 6], [9, 7], [8, 8], [8, 9], [8, 10], [7, 11], [7, 12], [7, 13], [7, 14], [8, 15]] |
| 3 | 21 | 7 | [[7, 21], [8, 20], [8, 19], [8, 18], [8, 17], [8, 16], [8, 15]] |
| 4 | 10 | 8 | [[8, 10], [7, 11], [7, 12], [7, 13], [7, 14], [8, 15]] |
| 5 | 15 | 11 | [[11, 15], [10, 15], [9, 15], [8, 15]] |
| 6 | 14 | 13 | [[13, 14], [12, 15], [11, 15], [10, 15], [9, 15], [8, 15]] |
| 7 | 10 | 10 | [[10, 10], [9, 11], [8, 12], [8, 13], [7, 14], [8, 15]] |
| 8 | 16 | 13 | [[13, 16], [12, 15], [11, 15], [10, 15], [9, 15], [8, 15]] |
| 9 | 20 | 11 | [[11, 20], [10, 19], [9, 18], [8, 17], [8, 16], [8, 15]] |
| 10 | 22 | 5 | [[5, 22], [6, 21], [7, 20], [8, 19], [8, 18], [8, 17], [8, 16], [8, 15]] |
| 11 | 19 | 3 | [[3, 19], [4, 19], [5, 18], [6, 17], [7, 16], [8, 15]] |
| 12 | 12 | 8 | [[8, 12], [8, 13], [7, 14], [8, 15]] |
| 13 | 19 | 10 | [[10, 19], [9, 18], [8, 17], [8, 16], [8, 15]] |
| 14 | 8 | 10 | [[10, 8], [9, 9], [8, 10], [7, 11], [7, 12], [7, 13], [7, 14], [8, 15]] |
| 15 | 12 | 10 | [[10, 12], [9, 13], [8, 14], [8, 15]] |
| 16 | 13 | 12 | [[12, 13], [11, 12], [10, 13], [9, 14], [8, 15]] |
| 17 | 15 | 6 | [[6, 15], [7, 14], [8, 15]] |
| 18 | 20 | 4 | [[4, 20], [5, 19], [6, 18], [7, 17], [8, 16], [8, 15]] |
| 19 | 8 | 5 | [[5, 8], [6, 9], [7, 10], [7, 11], [7, 12], [7, 13], [7, 14], [8, 15]] |
| 20 | 7 | 9 | [[9, 7], [8, 8], [8, 9], [8, 10], [7, 11], [7, 12], [7, 13], [7, 14], [8, 15]] |
| 21 | 15 | 3 | [[3, 15], [4, 15], [5, 15], [6, 15], [7, 14], [8, 15]] |
| 22 | 15 | 10 | [[10, 15], [9, 15], [8, 15]] |
| 23 | 10 | 12 | [[12, 10], [11, 11], [10, 12], [9, 13], [8, 14], [8, 15]] |
| 24 | 20 | 6 | [[6, 20], [7, 19], [8, 18], [8, 17], [8, 16], [8, 15]] |
| 25 | 13 | 11 | [[11, 13], [10, 13], [9, 14], [8, 15]] |
| 26 | 12 | 12 | [[12, 12], [11, 12], [10, 13], [9, 14], [8, 15]] |
| 27 | 15 | 9 | [[9, 15], [8, 15]] |
| 28 | 10 | 9 | [[9, 10], [8, 11], [7, 12], [7, 13], [7, 14], [8, 15]] |
| 29 | 8 | 7 | [[7, 8], [8, 9], [8, 10], [7, 11], [7, 12], [7, 13], [7, 14], [8, 15]] |
| 30 | 16 | 12 | [[12, 16], [11, 15], [10, 15], [9, 15], [8, 15]] |
| 31 | 12 | 7 | [[7, 12], [7, 13], [7, 14], [8, 15]] |

twenty-three scenarios satisfied the headed to the source condition. The accuracy of this UAV interface algorithm is evaluated as presented in Table 6.

It is essential to mention that the path steps generated by the RL agent resulted in 100% accuracy with a precision of one square meter, the cell size in the grid. Yet twenty-three scenarios out of thirty-one resulted from successful traversal to the gas source. The accuracy of the UAV interface algorithm obtained from this formula is 74.19%.

## The reasoning behind failed scenarios

The ROS simulator influences the performance of the UAV interface algorithm. It was required to generate yaw thrusts to change the directions according to the next step in the path of the UAV. The publisher in ROS can only receive a maximum of 45˚ of thrust angle for direction selection at one step. This is opposed to the maximum yaw thrust published in the algorithm, which is up to 180˚, both in positive and negative value. This opens another problem to either optimize the generated path for this specific simulator or introduce

**Table 6. Identification of failure conditions.**

| Scenario | Distance between UAV and Source Location (m) | Distance within 2.5 meters? | Headed to the Source Correctly? |
|---|---|---|---|
| 1 | 0.625 | YES | YES |
| 2 | 6.970 | NO | YES |
| 3 | 2.220 | YES | YES |
| 4 | 6.704 | NO | NO |
| 5 | 2.503 | YES | YES |
| 6 | 1.342 | YES | YES |
| 7 | 7.540 | NO | NO |
| 8 | 1.450 | YES | YES |
| 9 | 1.184 | YES | YES |
| 10 | 0.752 | YES | YES |
| 11 | 1.612 | YES | YES |
| 12 | 4.777 | NO | NO |
| 13 | 0.178 | YES | YES |
| 14 | 5.003 | NO | YES |
| 15 | 2.164 | YES | YES |
| 16 | 1.189 | YES | YES |
| 17 | 1.201 | YES | YES |
| 18 | 1.439 | YES | YES |
| 19 | 13.564 | NO | NO |
| 20 | 6.003 | NO | YES |
| 21 | 2.334 | YES | YES |
| 22 | 1.743 | YES | YES |
| 23 | 7.284 | NO | NO |
| 24 | 0.676 | YES | YES |
| 25 | 2.640 | NO | YES |
| 26 | 1.583 | YES | YES |
| 27 | 0.850 | YES | YES |
| 28 | 7.285 | NO | NO |
| 29 | 12.490 | NO | NO |
| 30 | 1.274 | YES | YES |
| 31 | 6.191 | NO | NO |

improvisation in the simulator for such an instruction on how to handle a swift turn of positive or negative 180˚.

The following figures show the trajectory obtained from 'rqt_multiplot' that has been superimposed with the background image of the environment illustration to visualize the trajectory of the Hector Quadrotor better. The source location is denoted with a white circle inside the figure. Fig 13 presents the trajectories towards the source mapped over the workspace illustration in which the red line indicates the trajectory of the UAV in ROS to reach the source location (in white) and the reasons behind and randomly selected failed and successful scenarios are presented as figures in Table 7.

## Work contributions

The key contributions of this work are listed below:

## Failed Scenarios

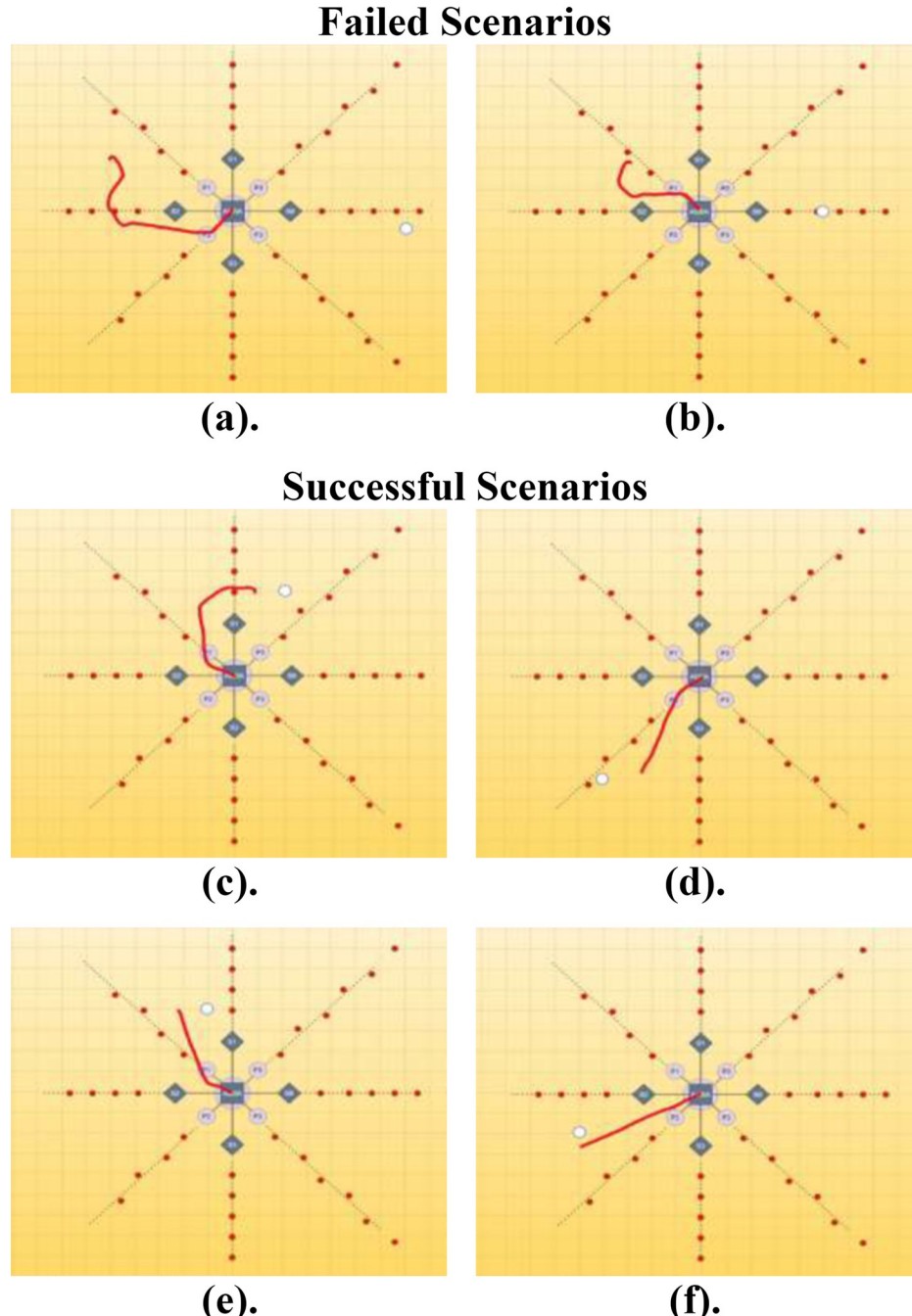

**Fig 13.** Trajectory illustrations: (a-b). Failed Scenarios, (c-f). Successful Scenarios.

i.  Designed and developed the Air Quality Sensor Array (AQSA) intended to be utilized as a Payload for UAVs (mobile sensors) based on MOX sensors.

ii.  Gas Dispersion Data collection from the fixed position sensors to study gas dispersion behavior and establish gas concentration maps to train AI agents.

iii.  Data Pre-processing and systematically generating a reward map for RL agents.

**Table 7.** Selected Failed (a-b) and Successful Scenarios (c-f) Explanations.

| Failed Scenarios | |
|---|---|
| (a) | In this scenario, the source point represented by a white circle is placed at (7, -1), and the final UAV location is at (-5.01, 2.428), which makes the distance between the two points 12.490 m. This scenario did not satisfy the required conditions. It is due to publishing the yaw thrust onto the UAV in simulation under ROS. |
| (b) | The white circle is placed at (5,0), and the final UAV location is at (-4.23, 5.201), making the distance between the two points 6.704 m. This scenario did not satisfy the required conditions well. It is also due to the above reasons that the yaw thrust in the simulator was not published correctly. |
| **Successful Scenarios** | |
| (c) | In this scenario, the source of emissions is randomly placed at (2,4), and the final UAV location is at (0.851, 3.693), making the distance between the two points 1.189 m. This scenario satisfied both conditions, *i.e.*, within 2.5 meters, and the heading angle is towards the source point. |
| (d) | Now the path generator is evaluated for the location at (-4, -5), and the final UAV location is at (-2.403, -4.781), with the distance between the two points being 1.612 m. This scenario also satisfied the necessary conditions as well. |
| (e) | The white circle is placed at (-1, 4), and the final UAV location is at (-2.272, 3.936), making the distance between the two points 1.274 m. This scenario also satisfied the conditions of distance and direction. |
| (f) | In this random test, the source was placed at (-5, -2), and the final UAV location was at (-5.002, -2.676), making the distance between the two points 0.676 m. It is one of the most accurate path estimation demonstrations to the source point. |

iv. Utilizing the idea of Difference of Arrival (DoA) of gas readings at multiple Metal Oxide (MOX) sensors, arranged at specific angles and positions, to intercept the plume source's distance and direction. It is a novel idea that reduces inherent inconsistencies of the sensors, adds redundancy and confidence in data, and provides wind vectors without using an anemometer.

v. We have developed a customized model-free learner policy for the RL agent specific for gas source localization applications. At the same time, there should be minimum path calculation time, from detecting the gas to estimating the emission point.

vi. Model-free RL agents are proposed to be utilized for autonomous path generations of UAVs.

vii. The scope of such agents is promising, where the sensor data can be obtained, and AI can be trained over the cloud. The learning policy, as a result, can be remotely transmitted to multiple field robots.

## Limitations

Following are some of the limitations of this work:

i. The experiments were conducted indoors where temperature, humidity, and wind vectors were known and almost consistent. For outdoor dispersion scenarios, it is recommended to collect new data.

ii. This work does not cover dynamic environmental conditions like wind gusts and varying temperatures.

iii. The sensing technology used for this work was Metal Oxide Semiconductor (MOX) sensors with certain limitations yet low-cost and readily available. There is a limited life span of each sensor before the degradation of its performance, and it should be replaced frequently.

iv.  The gas used for dispersion was butane ($C_4H_{10}$), heavier than air and can stay along the ground longer. Using a lighter gas than air would require sensors at multiple heights and locations. This work is limited to being tested for heavier gases.

v.  The gas used for dispersion experiments was stored in a pressurized container, weighing 280 grams, the impact of sudden release of gas molecules on the source location adds a limitation to this work, however, it had no significant impact on the path generation of the RL agent.

## Conclusion

A solution with a unique design of sensors array, mountable at the unmanned systems and powered by reinforcement learning, has been proposed in this work for gas source localization. A framework using an off-policy RL agent has been developed, enabling the UAV to navigate through a gas concentration map following its Q-table automatically. To attain this, a realistic reward map, based on the readings from the sensor array, was established to train the RL agent. The framework for reinforcement learning was designed based on the rewards that were calculated and mapped using realistic sensors-feed from the data collection phase. Wind vectors also influenced the rewards map, so to keep it constant, the experimentation for data collection was done in an indoor environment. To get the best performance from the reinforcement learning algorithm, the hyperparameters combinations for the reinforcement algorithm, such as epsilon, $\varepsilon$, discount rate, $\gamma$, and learning rate, $\alpha$, were evaluated, and the best combination for the fastest computational time is when $\varepsilon = 0.9$, $\gamma = 0.9$ and $\alpha = 0.9$.

When tested, the agent generated a path for unknown starting locations in the gas plume. The input from the testing data was given to the onboard sensors that generated paths autonomously with the help of its Q-table. The paths were further fed into a simulation environment in Robotic Operating System (ROS) to visualize the trajectory of Hector Quadrotor. ROS is known for its ability to require 'no change in codes' when needed to run on hardware. The UAV successfully navigated and reached the targets, and the gas source locations, with an accuracy of 74.19%. The fast computational time for data training and trajectory generation for automatic navigation was achieved by 1258.88 seconds for training and 6.2 milliseconds for path generation.

Based on its accuracy, computational time and performance results have shown enormous potential for reinforcement models for automatic UAV trajectory estimation in unknown environments. The strength of this work is that it opens a gateway to train machine learning algorithms for data collected through IOTs based sensors, and the off-line training can remotely transfer the Q-tables to heterogeneous robots equipped with the needed sensors.

## Supporting information

**S1 Data.**
(XLSX)

**S1 Graphical abstract.**
(TIF)

## Author Contributions

**Conceptualization:** Anees ul Husnain.

**Data curation:** Anees ul Husnain, Amirul Asyhraff Azmi.

**Formal analysis:** Anees ul Husnain, Amirul Asyhraff Azmi.

**Funding acquisition:** Masahiro Iwahashi.

**Investigation:** Norrima Mokhtar.

**Methodology:** Anees ul Husnain.

**Project administration:** Norrima Mokhtar.

**Resources:** Norrima Mokhtar, Masahiro Iwahashi.

**Software:** Anees ul Husnain, Amirul Asyhraff Azmi.

**Supervision:** Norrima Mokhtar, Noraisyah Binti Mohamed Shah, Mahidzal Bin Dahari.

**Visualization:** Anees ul Husnain, Amirul Asyhraff Azmi.

**Writing – original draft:** Anees ul Husnain.

**Writing – review & editing:** Norrima Mokhtar.

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
