## [Decision Letter · Decision Letter 0]

4 Jan 2023

PONE-D-22-22378Gas concentration mapping and source localization for environmental monitoring through unmanned aerial systems using model-free reinforcement learning agentsPLOS ONE

Dear Dr. Mokhtar,

Thank you for submitting your manuscript to PLOS ONE. After careful consideration, we feel that it has merit but does not fully meet PLOS ONE’s publication criteria as it currently stands. Therefore, we invite you to submit a revised version of the manuscript that addresses the points raised during the review process.

We look forward to receiving your revised manuscript.

Kind regards,

Mohd Nadhir Ab Wahab, Ph.D.

Academic Editor

PLOS ONE

Journal Requirements:

1 Please ensure that your manuscript meets PLOS ONE's style requirements, including those for file naming. The PLOS ONE style templates can be found at

“The study is supported by IIRG003(b)-19IISS, University of Malaya, Malaysia.”

4. Thank you for submitting the above manuscript to PLOS ONE. During our internal evaluation of the manuscript, we found significant text overlap between your submission and previous work in the [introduction, conclusion, etc.].

Please revise the manuscript to rephrase the duplicated text, cite your sources, and provide details as to how the current manuscript advances on previous work. Please note that further consideration is dependent on the submission of a manuscript that addresses these concerns about the overlap in text with published work.

[If the overlap is with the authors’ own works: Moreover, upon submission, authors must confirm that the manuscript, or any related manuscript, is not currently under consideration or accepted elsewhere. If related work has been submitted to PLOS ONE or elsewhere, authors must include a copy with the submitted article. Reviewers will be asked to comment on the overlap between related submissions (http://journals.plos.org/plosone/s/submission-guidelines#loc-related-manuscripts).]

We will carefully review your manuscript upon resubmission and further consideration of the manuscript is dependent on the text overlap being addressed in full. Please ensure that your revision is thorough as failure to address the concerns to our satisfaction may result in your submission not being considered further

Additional Editor Comments (if provided):

Please address the comments given by the reviewers.

Reviewers' comments:

Reviewer's Responses to Questions

**Comments to the Author**

1. Is the manuscript technically sound, and do the data support the conclusions?

Reviewer #1: Partly

Reviewer #2: Yes

2. Has the statistical analysis been performed appropriately and rigorously? 

Reviewer #1: No

Reviewer #2: Yes

3. Have the authors made all data underlying the findings in their manuscript fully available?

Reviewer #1: No

Reviewer #2: Yes

4. Is the manuscript presented in an intelligible fashion and written in standard English?

Reviewer #1: No

Reviewer #2: Yes

5. Review Comments to the Author

Reviewer #1: The abstract is too long. Some information could be inserted in the introduction chapter. This manuscript is not a project to have a challenges chapter or a goal one. This usually are included in the first chapter, meaning the Introduction. The presentation and the discuss of the findings must be concisely ( i.e. the goal was already presented in the abstract). The contribution is usually specified after the presentation of the measurement method and the results. The chapter called ”Reinforcement learning and gas concentration mapping” should be revised. It not clear. Table 4 should not be inserted in the text. The test were done in a controlled environment. What about the real conditions? The challenges that you mentioned in introduction how will influence you determinations? How many failed data will be then? In my opinion you should revised your manuscript to be an interesting one for the readers. Not a boring one. Specify briefly the present methods and their limits and you new one with benefits and limits.

Reviewer #2: The manuscript has been prepared in an excellent work. The data, results and discussion were briefly presented. Only a few queries; is this method applicable to all types of gas? Is there any limitation to apply this approach?

6. PLOS authors have the option to publish the peer review history of their article (what does this mean?). If published, this will include your full peer review and any attached files.

Reviewer #1: No

Reviewer #2: No

---

## [Author Response · Author response to Decision Letter 0]

20 Mar 2023

Please refer to the rebuttal letter, “Response to Reviewers.docx” for the corrections advised by the editorial office and reviewers.

---

## [Decision Letter · Decision Letter 1]

8 Oct 2023

PONE-D-22-22378R1Gas concentration mapping and source localization for environmental monitoring through unmanned aerial systems using model-free reinforcement learning agentsPLOS ONE

Dear Dr. Mokhtar,

Thank you for submitting your manuscript to PLOS ONE. After careful consideration, we feel that it has merit but does not fully meet PLOS ONE’s publication criteria as it currently stands. Therefore, we invite you to submit a revised version of the manuscript that addresses the points raised during the review process.

In general, reviewers have appreciated the work though one of the reviewers has suggested changes. Please improve the abstract by briefly mentioning the results achieved in a quantitative manner. Role of UAV in detection of gases mentioned on Line 69 could benefit from a reference 10.1109/ICCIS49240.2020.9257602. Similarly, autonomous behavior of a UAV on Line 74 can be mentioned with 10.1371/journal.pone.0282055. Moreover, with a website reference, please include date accessed. In Table 5, scenario 1 (row 1), please make sure that the closing brackets ] are consistent with opening brackets [. In the final version, please provide high resolution image files. 

We look forward to receiving your revised manuscript.

Kind regards,

Jamshed Iqbal, PhD

Academic Editor

PLOS ONE

Journal Requirements:

Reviewers' comments:

Reviewer's Responses to Questions

**Comments to the Author**

1. If the authors have adequately addressed your comments raised in a previous round of review and you feel that this manuscript is now acceptable for publication, you may indicate that here to bypass the “Comments to the Author” section, enter your conflict of interest statement in the “Confidential to Editor” section, and submit your "Accept" recommendation.

Reviewer #1: All comments have been addressed

Reviewer #3: (No Response)

2. Is the manuscript technically sound, and do the data support the conclusions?

Reviewer #1: Partly

Reviewer #3: Yes

3. Has the statistical analysis been performed appropriately and rigorously? 

Reviewer #1: Yes

Reviewer #3: N/A

4. Have the authors made all data underlying the findings in their manuscript fully available?

Reviewer #1: Yes

Reviewer #3: Yes

5. Is the manuscript presented in an intelligible fashion and written in standard English?

Reviewer #1: Yes

Reviewer #3: Yes

6. Review Comments to the Author

Reviewer #1: I appreciate you made the necessary correction and took action to exclude or to insert the specified requested chapters. The figures are now better explained. The manuscript is better composed and have the necessary parts. The responses gave to the reviewers are clear and the authors made the effort to present their work properly.

Reviewer #3: I will start my comments with the appreciation of detailed work that was required to put together this manuscript. Overall, it is a thorough work within the mentioned limitations about the sensor array and data acquisition locations. Following are a couple of minor adjustments that can make this work better in terms of usability and reproducibility of results.

1- A discussion on sensor type is needed. the reason behind that is the efficiency and selectivity of butane sensors is highly dependent on sensor types. Was it a high temperature sensor or a low temperature sensor? what was the response time of this sensor for a reliable reading? what was the calibration procedure? what will be impact of temperature variations on the performance of this sensor?

2- Along the same lines, it will be nice to add whatever was not done from the above-mentioned steps in the limitations or future work sections.

3- Another important discussion to be added in data acquisition section and/or limitations section is the impact of gas plumes emitted by a pressurized container. The gas temperature will be significantly different as it comes out of the pressure vessel. Hence the sensor readings may report a concentration value that is not a true indicative of a source location. Special adjustments will be required to calibrate the sensor array against this anomaly.

4- And finally include any and all of the above data sets to the submitted data. As an example, you should include the calibration factors of all sensors that you used.

7. PLOS authors have the option to publish the peer review history of their article (what does this mean?). If published, this will include your full peer review and any attached files.

Reviewer #1: No

Reviewer #3: No

---

## [Author Response · Author response to Decision Letter 1]

22 Nov 2023

Response to the reviewers has been attached with this re-submission.

---

## [Editor Report · Decision Letter 2]

27 Dec 2023

Gas concentration mapping and source localization for environmental monitoring through unmanned aerial systems using model-free reinforcement learning agents

PONE-D-22-22378R2

Dear Dr. Mokhtar,

We’re pleased to inform you that your manuscript has been judged scientifically suitable for publication and will be formally accepted for publication once it meets all outstanding technical requirements.

Kind regards,

Jamshed Iqbal, PhD

Academic Editor

PLOS ONE
---

## [Editor Report · Acceptance letter]

14 Feb 2024

PONE-D-22-22378R2 

PLOS ONE

Dear Dr. Mokhtar, 

I'm pleased to inform you that your manuscript has been deemed suitable for publication in PLOS ONE. Congratulations! Your manuscript is now being handed over to our production team.

Kind regards, 

on behalf of

Dr. Jamshed Iqbal 

Academic Editor

PLOS ONE